

# Stream laws in tectonic landscape analogue models

Riccardo Reitano[1], Romano Clementucci[1], Ethan M. Conrad[2], Fabio Corbi[3], Riccardo Lanari[4], Claudio Faccenna[1,5], Chiara Bazzucchi[1]

[1]Departmenbt of Science, University of Rome "Roma TRE", Laboratory of Experimental Tectonics, Rome, Italy
[2]Department of Geological Sciences, Jackson School of Geosciences, The University of Texas at Austin, Austin, TX, USA
[3]National Research Council - CNR, Istituto di Geologia Ambientale e Geoingegneria, Italy
[4]Department of Earth Science, University of Florence, Florence, Italy
[5]Lithosphere Dynamics, Helmholtz Centre Potsdam, German Research Centre for Geosciences (GFZ), Potsdam, Germany

*Correspondence to*: Riccardo Reitano (r.reitano@gmail.com)

*Keywords*: Tectonic Geomorphology, Erosional laws, Analogue modelling.

*Keypoints*: Effect of boundary conditions on analogue models using water-saturated granular materials; channelization over more diffusive processes; application of channel power law on analogue models; comparison between natural and analogue geomorphic features.

## Abstract

The interplay between tectonics and surface processes defines the evolution of mountain belts. However, correlating these processes through the evolution of natural orogens represents a scientific challenge. Analogue models can be used for analyzing and interpreting the effect of such interaction. We use nine analog models characterized by different combinations of imposed regional slope and rainfall rates to investigate how a small scale orogen evolves in response to tectonics and surface processes. We show how the combination of these parameters control the development of drainage networks and erosional processes. We quantify the different morphological expression of analogue landscapes in terms of a proposed *Ae* number that accounts for both observables and boundary conditions. We find few differences between analogue models and natural prototypes, in terms of parametrization of the detachment-limited stream power law. We observe a threshold in the development of channelization, modulated by a tradeoff between applied boundary conditions.

## 1    Introduction

Accurately interpreting the continuous interaction between tectonics and surface processes in mountain belts is one of the main challenges that geologists have faced in the last century. Many limitations exist due to the different spatial and temporal scale depths in which crustal and mantle processes impact the surface. Thus, it is difficult to univocally interpret how different factors interact to build the present-day landscape. Analog models allow for a useful direct control on the evolution of the studied physical process (*e.g.*, Reber et al., 2020), overcoming many of these limitations. Tectonics, erosion, and sedimentation play an integrated role in the evolution of mountain belts with complex and poorly constrained feedbacks. During the last decades modelers analyzed these feedbacks, from the rejuvenation of streams (*e.g.*, Schumm and Parker, 1973; Schumm and Rea, 1995) to the more complex evolution of whole orogenic systems (*e.g.*, Bonnet, 2009; Graveleau and Dominguez, 2008; Guerit et al., 2016; Lague et al., 2003; Tejedor et al., 2017; Viaplana-Muzas et al., 2019; Reitano et al., 2022). However, all the previous analog modeling efforts are based on the robustness of the characterization of material used in the experiments (e.g., Graveleau et al., 2011; Reitano et al., 2020) and on the scaling to natural prototypes (*e.g.*, Graveleau et al., 2011; Paola et al., 2009). While the mechanical, frictional





and erosional properties have been characterized empirically and analytically by different authors (*e.g.*,
Bonnet and Crave, 2003; Lague et al., 2003; Graveleau et al., 2011; Reitano et al., 2020), the application
of erosion laws to analog systems is still a matter of debate. In particular, a definition of the analogue
materials response to the applied boundary conditions and an understanding of the variables that modify
this response are still missing. These characterizations become even more important considering that a
perfect scaling between natural and experimental flow laws is currently missing, limiting the reach of
analog studies to insights derived from qualitative process similarity (Paola et al., 2009). In this work
we analyze how different boundary conditions affect the evolution of analogue landscapes, in terms of
tectonics (imposed regional slope) and climate (rainfall rate). The methodologies implemented here
allow us to isolate how different components control features like channelization, morphometrics and
incision rates. We are able to define the ranges of external forcings (*i.e.*, rainfall rate and imposed
regional slope) in which different morphological features develop (*e.g.*, channels or diffusive
processes). Finally, we discuss if and how the erosional parametrization implemented in nature (stream
power law) apply to analogue models. Analog models performed in this study represent slow tectonic
regions (*e.g.*, very low uplift/erosion rate such as the Anti-Atlas of Morocco; Lanari et al., 2022;
Clementucci, 2022), or passive margins such as the Western Ghats escarpment in the western Arabian
Sea (Wang and Willett, 2019) and the Southern Australian Escarpment (Godard et al., 2019).
## 2      Experimental Setup, scaling, and erosional laws
The analogue material is a granular, water saturated mixture made of 40 wt.% of silica powder, 40 wt.%
of glass microbeads and 20 wt.% of PVC powder (Reitano et al., 2020). We fill a $35\times30\times5$ cm$^3$ box
with the material and place it on a reclinable table (Fig. S1). The rainfall system is made of nozzles that
provide a dense fog to trigger surface processes. The droplet size in the fog is lower than 100 µm to
avoid rainsplash erosion (e.g., Graveleau et al., 2012). We apply three different rainfall rates on the
models: 9, 22, and 70 mm h$^{-1}$ by controlling the number of implemented nozzles. The angle between
the table and the horizontal (i.e., imposed regional slope) is also fixed at three different values, 10°, 15°,
and 20°. Both rainfall rate and imposed regional slope are kept constant throughout the experiment. In
total, we investigate nine different imposed regional slope-rainfall configurations (Table S1). We use a
camera to capture top view digital images of the experiment evolution, and a high-resolution laser
scanner to acquire DEMs at defined time-steps. Vertical and horizontal resolution of the laser are 0.07
mm and 0.05 mm, respectively.
Considering a length scaling factor $L^* = 10^{-5}/10^{-6}$ (1 cm = 1-10 km), a gravitation acceleration scaling
factor $g^* = 1$, and a velocity scaling factor $v^* = 10^4$-$10^5$ (1 cm h$^{-1}$ = 0.1-1 cm yr$^{-1}$), the time scaling
factor $t^*$ is computed by (Reitano et al. 2020; Reitano et a., 2022)
$$t^* = \frac{4L^*}{v^*}$$  Eq. (1)
, such that 1 min in the models corresponds to 5 to 50 kyr in nature.
The change in elevation (*dz*) over time (*dt*) of a point on the surface results from the competition
between the rock uplift rate $U$ and the erosion or sedimentation rate $E$
$$\frac{dz}{dt} = U - E$$  Eq. (2)
For fluvial networks that show exposed bedrock at the base, the erosion rate is typically expressed as a
function of a "detachment-limited stream power law" (*e.g.*, Goren et al., 2014a; Goren et al., 2014b;
Howard, 1994; Howard and Kerby, 1983; Tucker and Whipple, 2002)
$$E = \kappa Q^m S^n$$  Eq. (3)
, where $\kappa$ is the bedrock erodibility, $Q$ is the channel discharge, $S$ is the channel slope, and $m$ and $n$ are
positive exponents accounting for channel geometry, basin hydrology and erosion processes. Since $Q$
is function of drainage area $A$ and rainfall rate $P$ ($Q = A \cdot P$), Eq. (3) can be rewritten as



$E = KA^m S^n$                                                                                    Eq. (4)
, where $K = \kappa \cdot P^m$ is controlled by bedrock lithology, incision process and climate (here rainfall rate).
Assuming steady-state condition, graded fluvial channels are correctly described by a power
relationship between channel slope $S$ and drainage area $A$ (Flint, 1974)
$S = k_s A^{-\theta}$                                                                              Eq. (5)
, where $k_s$ and $\theta$ are the channel steepness and the concavity indexes, respectively (e.g. Lanari et al.,
2020; Sembroni et al., 2016; Wobus et al., 2006, Kirby and Whipple, 2012; Hack, 1957, 1960).
Merging Eq. (4) and (5), and considering that $\theta = m/n$, we can express the result in a logarithmic form
$\log_{10} E = n\log_{10} k_s + \log_{10} K$                                                      Eq. (6)
Eq. (6) is thus the equation of a line ($y = mx + q$), where $n$ and $\log_{10}K$ are the slope and the intercept of
the line, respectively.
From the DEMs we compute: (i) topographic metrics such as basin slope and local relief; (ii) channels
metrics ($k_s$ and $\theta$) and channel profiles; (iii) eroded volumes, incision rate and erosion maps. We
describe the methodologies used for extracting the eroded volumes and incision rates in the
Supplementary Information. The above-mentioned analysis are performed using *ad hoc* MATLAB
scripts (see data repository) and the TopoToolbox package (Schwanghart and Scherler, 2014).
**3      Results**
Table S1 shows a list of the performed experiments, where the first and last two digits of the models'
name represent the imposed regional slope and the imposed rainfall rate, respectively (*e.g.*, mod1522,
imposed regional slope 15° and rainfall rate 22 mm h⁻¹). These model runs are interpreted highlighting
the effect of boundary conditions on the geomorphic models' evolution. Results refer to ten time steps
that highlight key stages in the evolution of each model (Supplementary Information), or to the entire
experimental run (300 min). Increasing the imposed regional slope and the imposed rainfall rate (from
mod1009 to mod2070), the maximum surface slope (MSS) increases systematically from 30° to 80°
(between 25th and 75th percentile, Fig. 1a). Mod1009 shows the highest distribution of values.
Conversely, the surface slope mode (SSM) decreases for models with the same regional slope but
different rainfall rates (*e.g.*, mod1509, mod1522, mod1570), except for mod1009 and mod1022 (Fig.
1b), whereas the mode increases for models with the same rainfall rate but different imposed regional
slopes (*e.g.*, mod1009, mod1509, mod2009). Models with rainfall rates of 70 mm h⁻¹ show the lowest
values in SSM, yet the broadest range of values (between 5° and 15° from mod1070 to mod2070). The
decreasing trend in the SSM is also observed in the morphological expression of the model's surface
(Figs. S2 and S3). The models with a rainfall rate up to 22 mm h⁻¹ show clear channelization during the
final stage of the evolution, while models with rainfall rates equal to 70 mm h⁻¹ shown little to no
channelization at final the stage. Only mod2070 develops well-defined channels (branching, narrow
channels, incision focused on valleys and into channels). The concavity index ranges from 0.1 to 0.4
(between 25th and 75th percentile). Models with the highest rainfall rate typically show the highest
concavity values with correspondingly, the highest variability (mod1070, mod1570, mod2070, Fig. 1c).
The local relief was extracted using a moving window with a radius of 10 mm. The maximum local
relief (MLR) and local relief mode (LRM) show a similar pattern with respect to the surface slope (Fig.
1d, e). The MLR increases from mod1009 to mod2070, while the LRM decreases for models having
rainfall rate equals to 70 mm h⁻¹ with respect to models having the same imposed regional slope but
different rainfall rate. The same information can be deduced by visual inspection of DEMs (Figs. S2,
S3).
The amount of eroded material ranges from $0.2 \cdot 10^6$ (mod 1009) to $2.3 \cdot 10^6$ mm³ (mod 2070) at the last
stage of the experiments (Fig. 2a). Both increased rainfall rate (e.g., mod1009, mod1022, mod1070)
and increased imposed regional slope (*e.g.*, mod1022, mod1522, mod2022) result in higher amounts of





eroded material. All models show an initial phase where the material flux is highest with a later phase
of decline (Reitano et al., 2020), tending toward stability. For example, in mod2070, 60 min of
experimental time are enough to erode 1.4 mm$^3$ of material, while in the next 240 min only additional
$0.9 \cdot 10^6$ mm$^3$ of material are eroded. This different behavior is most apparent in models with high rainfall
rates ($\geq$ 22 mm h$^{-1}$) and slopes ($\geq$ 15°). The maximum incision rate increases with the imposed regional
slope and rainfall rate (Fig. 2b), from < 5 mm h$^{-1}$ (mod1009) to ca. 55 mm h$^{-1}$ (mod2070).
We extract values for $E$ and $k_s$ of four main channels for each time step and model (40 channels per
model, total = 360). Despite the low R$^2$ values (0.01 – 0.28), $n$ ranges between -0.18 and 0.14 with $K$
values between 0.77 and 23.51 mm$^{1-2m}$ h$^{-1}$. Values of $K$ increase as a function of the slope and rainfall
rate, while $n$ does not show a clear trend. On the right hand of Fig. 3, we plot data for models with the
same imposed regional slope but different rainfall rates. Models with the same regional slope show a
gradually increasing $k_s$ and incision rate in response to increased rainfall rates (Fig. 3b). Interestingly,
estimates of $K$ gradually increase in models with imposed regional slope of 10° to 20° (Fig. 3b).
## 4        Discussion
### 4.1      Type of erosion as a function of boundary conditions
Our analogue models are controlled exclusively by imposed regional slope and rainfall rate, as no other
external forcing is applied (e.g., vertical uplift or horizontal advection of material). Higher rainfall rates
(70 mm h$^{-1}$ in this work) tend to inhibit the development of a channelized and branching channel
network in favor of more diffusive and mass wasting processes. This trend can be deduced by analyzing
the DEMs of mod1070 and mod1570 (Fig. S2), or simply by noting the diffuse nature of erosion under
high rainfall conditions (Fig. S3). Mod2070 develop channelization more than mod1070 and mod1570
but, similarly, mod1570 show slightly more channelization than mod1070. These observations suggest
that the higher the slope, the more effectively a system responds to the high rainfall rates in terms of
channelization. Considering the relationship between channelization and boundary conditions, the
results of our experiments suggest that low imposed regional slope (mod1009, mod1022) or low rainfall
rate with average imposed regional slope (mod1509) result in a channelization characterized by low
incision values (<25-30 mm at final stage). It thus appears that a threshold exists in rainfall rate for the
developing of channelization, modulated by the slope over which erosion acts. For example, a rate of
70 mm h$^{-1}$ (mod1070 and mod1570) is too high for a proper channel network to develop, while a higher
imposed regional slope (mod2070), provide sufficient potential energy to the system to develop
channelization (*e.g.*, Burbank and Anderson, 2012). Thus, the tradeoff between imposed regional slope
and rainfall rate controls the development of channelization. For higher rainfall rate (mod1070 and
mod1570), a sheetlike runoff lowers the model slope homogeneously (Fig. 1b). Furthermore, both the
SSM and the LRM drop with respect to models with the same imposed regional slope but lower rainfall
rate. From these observations, we argue that at high rainfall rate channelization is subordinate to
diffusive processes (controlled by ridge stability) at final stages of model evolution.
In landscapes where incision is function of the detachment of particles from the riverbed, the erosion
rate is proportional to the shear stress (*e.g.*, Whipple and Tucker, 1999; Yanites et al., 2010). Higher
rainfall rate (*i.e.*, higher water discharge) increases the effective shear stress on riverbed (Thoman and
Niezgoda, 2008). Since water discharge increases also the channel width (*e.g.*, Shibata and Ito, 2014;
C. W. Wobus et al., 2006), for high water discharges (*i.e.*, rainfall rate) the shear stress can distribute
over time over wide and flat surfaces instead of focusing in valleys (Lamb et al., 2015 and references
therein). At high rainfall rate our models show incipient channelization in the initial stages.
Channelization is lost in favor of more diffusive processes during late stages of model evolution. High
rainfall rate (*i.e.*, high water discharge) lead to higher sediment supply, that can widen channels





(Finnegan et al., 2007; Johnson and Whipple, 2010), eventually erasing their morphological expression.
Finally, we must address the erosional threshold defined in the works of Lague et al. (2003) and
Graveleau et al. (2011). This threshold must be overcome before significant erosion and transport
occurs, and specifically apply to models at low imposed regional slope, which may thus lead to even
less channelization. Interestingly, effective channelization does not affect the volume of eroded
material, which increases with the imposed regional slope and rainfall rate (*e.g.*, the erosion flux is
higher in mod1570 than mod1522 and mod1070). To quantify the relationship between the incision
rate, the eroded volume, and imposed regional slope and rainfall rate, we define a dimensionless number
for erosion $Ae$, defined as
$$Ae = \frac{I}{V} V_n \frac{S_e}{R} \qquad\qquad\qquad \text{Eq. (7)}$$
where $I$ is the incision rate (mm h$^{-1}$), $V$ is the eroded volume, $V_n$ is the eroded volume normalized with
respect to all the models, $S_e$ is the imposed regional slope and $R$ is the applied rainfall rate. We plot the
$Ae$ as a function of time (Fig. 2c) and at 300 min (end of the experimental run) over the ratio between
imposed regional slope and rainfall rate ($S_e/R$, Fig. 2d). Nearly every model shows a similar evolution
of $Ae$ number in time, except for those with medium to high imposed regional slope and high rainfall
rates ($S_e > 15º$, $R = 70$ mm h$^{-1}$, mod1570 and mod2070). The latter experiments attain higher values of
$Ae$ number throughout the whole experimental run (even showing a decrease in time). The decrease of
$Ae$ for mod1570 and mod2070 is due to the decrease in the volume erosion rate (Fig. 2a). The same
decrease affects all the other models, but it is less clear due to the lower $Ae$ numbers. Fig. 1d clearly
shows the relationship between the imposed boundary conditions ($S_e/R$) and their effect on the models'
evolution ($Ae$ number). Models tend to align on three straight lines (Fig. 2d), following the rainfall rate.
When $Ae$ number is higher than $1 \cdot 10^{-3}$ and $S_e/R$ is lower than 0.5 (dark gray, Fig. 1d), channelization is
less important than diffusive processes in controlling the erosion. Models with high rainfall rate
(mod1070, mod1570, mod2070) or low rainfall rate and low imposed regional slope (mod1009) show
low channelization with respect to the other models. Even if mod2070 and mod1009 show
channelization in their DEMs (Fig. S2), the level of incision (Fig. S3) shows that erosion is broadly
distributed, and not focused in channels or valleys. When $Ae$ number is lower than $1.5 \cdot 10^{-3}$ and $S_e/R$ is
lower than 1 (Fig. 1d, medium light gray), channelization processes are present and are responsible for
the erosion of the surface. Still, channels and valleys are wide (ca. 6 cm) with respect to models where
$Ae$ number is lower than $1 \cdot 10^{-3}$ and $S_e/R$ is higher than 1 (mod1009, mod1509, mod2009, light gray in
Fig. 1d). In these latter cases, the maximum incision rate is lower than 20 mm h$^{-1}$ (Fig. 1b), and the
eroded volumes range between 0.1 and $0.75 \cdot 10^6$ mm$^3$.
It is then clear that a threshold for developing channelization exists. This threshold is modulated mainly
by the rainfall rate, but also by the imposed regional slope. For instance, models with $Ae$ number lower
than $1.5 \cdot 10^{-3}$ and $S_e/R$ higher than 0.5, develop channelization. We propose that the $Ae$ number can be
used as a threshold parameter to define if and how channelization processes may develop in analogue
models.
**4.2    Geomorphological metrics in analogue models and natural prototypes**
The concavity index $\theta$ of the selected channels (for every model) is usually lower than 0.4 (Fig. 1c,
values between 25$^{th}$ and 75$^{th}$ percentile). The lower values of $\theta$ are related to straight longitudinal
channel profiles (Fig. S4) that are established over the course of a model (Duvall, 2004; Reitano et al.,
2020; Whipple and Tucker, 1999). Still, $\theta$ is comparable between analogue models and natural
prototypes (within 0.1, Reitano et al., 2020). However, we do not normalize the steepness index $k_s$ by a
reference concavity index ($k_{sn}$), as usually done in literature (*e.g.*, Cyr et al., 2010; DiBiase et al., 2010;
Kirby and Whipple, 2012; Lanari et al., 2020; Tucker and Whipple, 2002). This approach is used so





that the steepness index directly reflects channel characteristics, without modifying the data with a
normalization parameter.
If $n = 1$, there exists a linear relationship between incision rate and $k_s$ - i.e., an increase in the steepness
index increases the incision rate and vice versa (Eq. 6). Previous works that computed values of $n$ for
different natural landscapes show that $n$ is typically greater than or equals 1 (*e.g.*, DiBiase et al., 2010;
Harel et al., 2016; Ouimet et al., 2009), meaning  that incision rates are extremely sensitive to variation
in the $k_s$ of rivers in active tectonic settings (Kirby and Whipple, 2012). In slow tectonic settings, the
incision rates show a lower sensitivity to the steepness index (Olivetti et al., 2016; Clementucci, 2022).
For this case, we show that $n$ has values generally lower than 1, even negative (Fig. 3, left plots). Since
$n$ is dimensionless, a 1:1 comparison between models and nature is possible. The $n$ results from our
analogue models, are closer to estimates of $n$ values from slow tectonic settings compared to active
domains, where $n$ is greater than 1 (Figs. 3b and 4; Kirby and Whipple, 2012; Clementucci, 2022).
Similar to slow tectonics landscapes in nature, incision rates of analogue models are less sensitive to
variations in the steepness index $k_s$. Moreover, the low $R^2$ (Fig. 3) testifies a poor relationship between
incision rate and $k_s$ ($n < 1$). Nevertheless, we observe a trend in the relationship between incision rate
and $k_s$ as a function of the applied rainfall rate (Fig. 3, right panel). For higher rainfall rates, the incision
rate increases as expected, but only with a slight increase in the $k_s$. This consideration supports the fact
that the $k_s$ plays a minor role in controlling the incision rate, like slow tectonic natural domains. Note,
many works in literature compute values of incision rate and $k_s$ (or $k_{sn}$) for tectonically active regions.
The models presented here are similar to natural prototypes where tectonics is absent or subordinate
with respect to surface processes. To show this, we collect values of incision rate and $k_{sn}$ for natural
prototypes that show no or almost no active tectonics, and compute values of $n$ and $K$ (Fig. 4). Estimates
of $n$ from the analogue models present a good match with data from the selected slow tectonics natural
prototypes (Fig. 4), showing values between -0.06 and 0.32 for analogue models (Fig. 3) and -0.12 and
0.62 for natural systems (Fig. 4). This relationship is apparent despite the low $R^2$ value from linear
regression of data from analogue and natural cases. Importantly, the difference in the range of $n$ values
can be likely decreased by excluding data with a higher uplift rate in the slow tectonic settings (*e.g.*,
granite dominated basins; Clementucci, 2022).
Values of $K$ are more difficult to compare between models and nature, due to the complex dimensions
that characterize the parameter ($mm^{1-2m} h^{-1}$), which require not only the application of a scaling factor,
but the definition and comparison of the $m$ variable. Still, it is possible to analyze how the boundary
conditions affect $K$. Because $K$ is mainly function of the lithology and climate (section 2), and we use
the same homogenous material for all the models, $K$ only highlights the effect of the rainfall rate. There
is a nearly perfect linearity between $K$ and the rainfall rate. For example, from 9 to 22 and from 22 to
70 mm h-1, the rainfall rate increases 2.4 and 3.18 times, respectively. This is also apparent for $K$, with
the only exception of the experiment with both moderate imposed slope and rainfall rate (mod1522).
Interestingly, there is also a linear relationship between $K$ and the slope. For example, in Fig. 3 when
moving from high to low imposed slope. Thus, we speculate that $K$ is not only function of lithologies
and climate, but also of the regional slope and, consequently, of the integrated topographic response to
tectonic rates, as observed for the natural prototypes (Peifer et al., 2021).
**5      Conclusions**
We investigate the role of boundary conditions (imposed regional slope and rainfall rate) on the
morphological evolution of nine different analogue models. The models systematically test various
configurations of boundary conditions. We analyze how the stream power law used in natural
landscapes apply to analogue models. We find that a threshold exists for the development of
channelization in terms of boundary conditions ($Ae$ number and $S_e/R$ ratio), and that this threshold is



the result of a tradeoff between the imposed regional slope (tectonics) and the rainfall rate (climate).
We find that combining imposed regional slope and rainfall rate result in three possible results:
• $Ae$ number higher than $1 \cdot 10^{-3}$ and $S_e/R$ lower than 0.5: diffusive processes are dominant with
respect to channelization (mod1070, mod1570, mod2070);
• $Ae$ number lower than $1.5 \cdot 10^{-3}$ and $S_e/R$ lower than 1: channelization controls the erosion of the
surface, with wide channels and valleys (ca. 6 cm);
• $Ae$ number lower than $1 \cdot 10^{-3}$ and $S_e/R$ higher than 1 (mod1009, mod1509, mod2009):
channelization processes are the main characters in controlling the morphologies of the
analogue landscapes.
Summarizing, high rainfall rate (70 mm h$^{-1}$) inhibits the development of channels, but not if coupled
with high imposed regional slope (20°). Similarly, low imposed regional slope and low rainfall rate (10°
and 9 to 22 mm h$^{-1}$, respectively) develop channelization characterized by low incision. When looking
at the parametrization of the stream power law for erosion, we find that analogue models are described
by parameters which values slightly differs from the natural prototypes (*e.g.*, $n$ exponent), and by
parameters that are tuned by applied boundary conditions imposed regional slope and rainfall rate (*e.g.*,
$K$ constant). Given these findings, we propose that analogue models here presented can and must be
used in the interpretation of the interaction between tectonic and surface processes, taking into account
the limitations discussed here and the range of applicability of the boundary conditions.
**Data Availability**
Data have been uploaded open access on GFZ Data Service: https://dataservices.gfz-
potsdam.de/panmetaworks/review/08e477e94c543368eec875408be0db5a4e08ff87ac66f5f03736fcd97
6b96ac0/. These data will receive a doi prior to publication.
**Author contribution**
RR, RC proposed the original idea. RR, RC and EMC designed the experiments and RR carried them
out. RR and FC developed the codes for the model analysis that has been performed by all authors.
Interpretation of results, writing, reviewing and editing were performed by all authors.
**Acknowledgments**
In Figures 2, S2 and S2 we used the perceptually uniform colormap, Roma and Davos, by Fabio
Crameri. The grant to the Department of Science, Roma Tre University (MIUR-Italy Dipartimenti di
Eccellenza, ARTICOLO 1, COMMI 314-337, LEGGE 232/2016) is gratefully acknowledged.

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

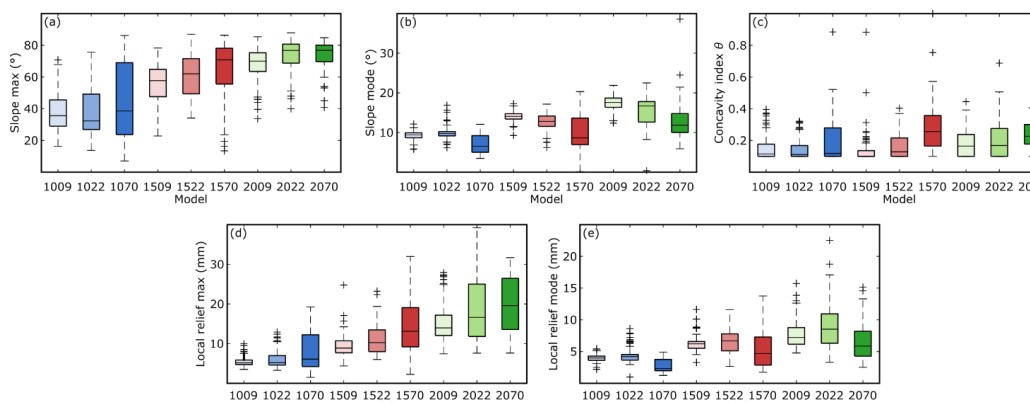


**Figure 1. Geomorphological and channels metrics of the performed experiments. The black solid lines indicate the median, while the bottom and top edges of the colored boxes indicate the 25th and 75th percentiles, respectively. The black whiskers outside the boxes cover the data point at <25th and >75th percentiles that are not considered outliers, here indicated by black crosses. The color saturation in the boxes is related to the applied rainfall rate (less saturated, less rainfall rate). The blue, red and green boxes refer to models at 10°, 15° and 20° of imposed slope, respectively.**

440

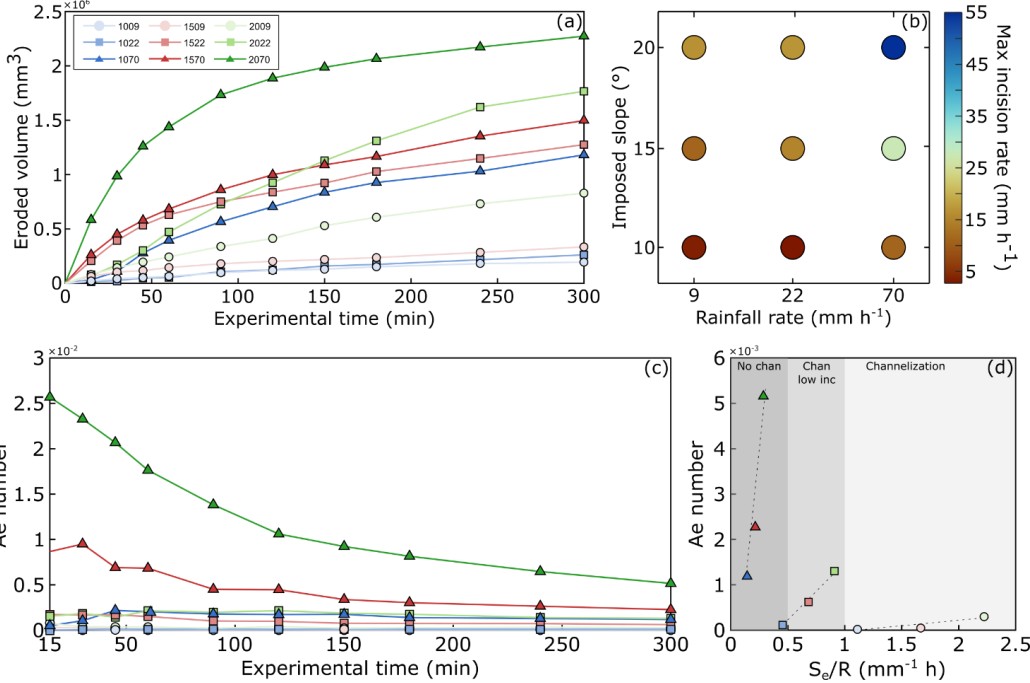

**Figure 2. (a) Amount of eroded material for the performed experiments (the color coding follows Fig. 1); (b) Maximum incision rate as a function of the imposed boundary conditions; (c) Ae number computed for all the models; (d) Ae number plotted over the ratio between the imposed slope ($S$) and the applied rainfall rate ($R$).**




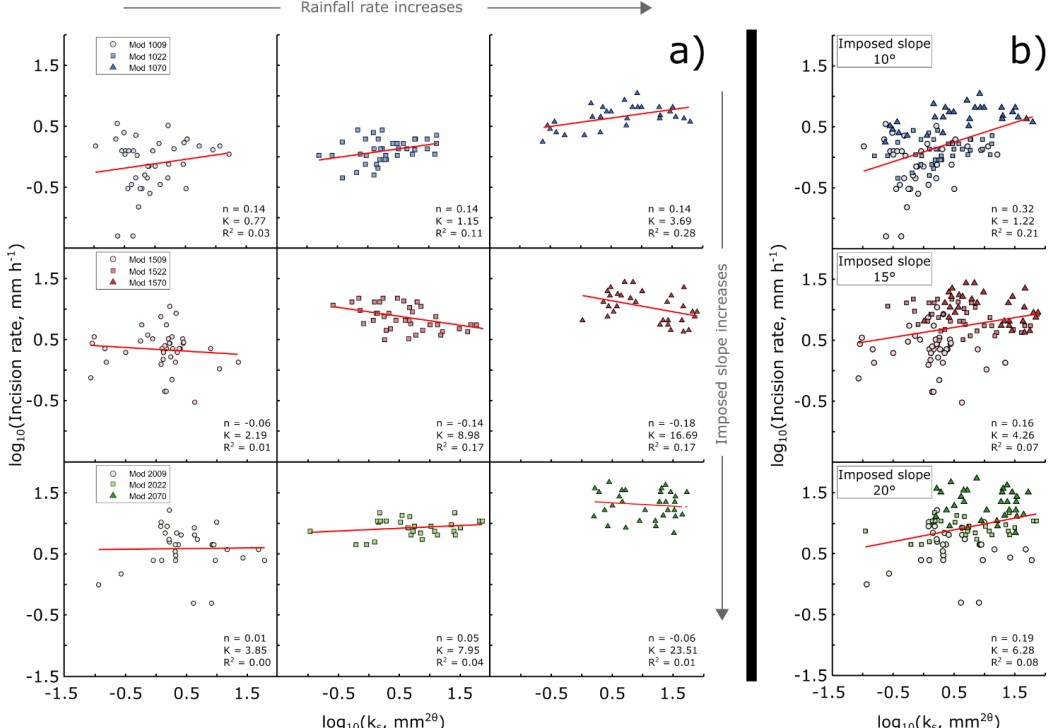

**Figure 3. a) logarithm of the incision rate over the logarithm of the steepness index $k_s$ for all the models. Imposed regional slope increases from top to bottom. Rainfall rate increases from left to right. Every plot shows four channels at every time step, forty channels total (colored dots). The linear regression is shown by the red line. Values related to the linear regression ($n$ and $K$) are shown in the bottom right corner of every plot, together with the $R^2$ (units for K are mm$^{1-2m}$ h$^{-1}$); b) same data of a), but plotted for every slope, without differentiating for rainfall rate.**

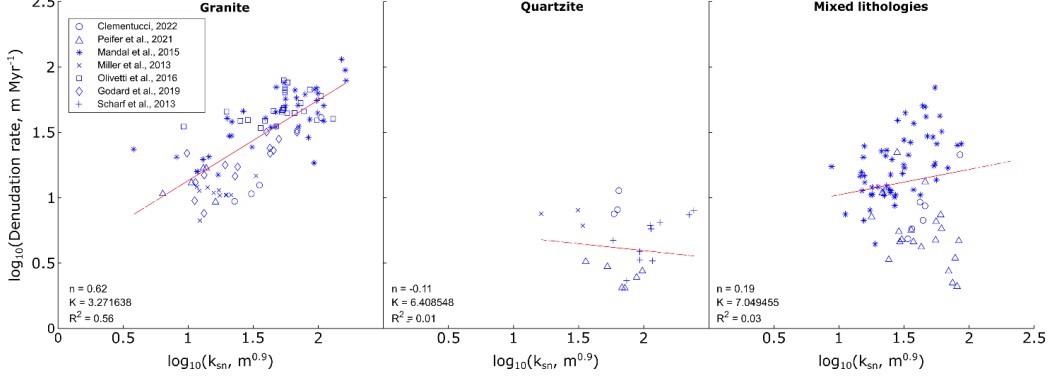

**Figure 4. Logarithm of the incision rate over the logarithm of the steepness index $k_s$ for the selected natural prototypes. The linear regression is shown by the red line. Values related to the linear regression ($n$ and $K$) are shown in every plot, together with the $R^2$.**