# Peer review of "Stream laws in tectonic landscape analogue models"

_EGUsphere, 2022_

## Referee Comment (RC1)

**General Comments:**

This manuscript builds on previous work (Reitano et al., 2020) to explore the evolution of drainages in response varying precipitation rates and regional slopes using analog experiments. The manuscript is well-structured, and poses an analog model configuration that can be used to investigate landscape evolution in future studies. Furthermore, the authors rely on well-established channel profile analyses to quantify the relationship between analog models and natural settings.

This work presents a good contribution to the field geomorphology, especially as we try to link models with nature. However, I have two overarching critiques of the manuscript that should be addressed by the authors. First, there appears a lack of clarity in some of the methodology. Some metrics are given and discussed without an explanation of how they were derived (see notes given in the Specific Comments section). Explaining these methods in further detail will help give robustness to the manuscript results and interpretations. Second, I think there is a disconnect between the model setup and its relationship to tectonic orogens. These models do not impose uplift rates, but instead start with a background slope; essentially simulating an orogen that rapidly grew and stopped, with erosion occurring only after the tectonic phase. The authors rightly state that they're models fall more in line with passive margin settings, but also imply the background slopes are proxy for tectonics. I think this needs some further justification.

**Specific Comments:**

Line 15 – 16: Authors argue for the need to adequately model tectonic settings, but their setup simulates low uplift rates / passive margins. This motivation seems slightly skewed.

Line 48: Here and elsewhere in the paper, 'boundary condition' seems to be used interchangeably with model forcing; these are not the same. Rainfall rate and imposed slope should be described only as model forcings (and even then, the imposed slope is more of an initial condition).

Line 63: Authors don't change nozzle flow rate to test precipitation rates, but instead change the number of nozzles. This should be elaborated on – what configurations were used and what tests conducted to ensure uniform rainfall?

Line 84: Some studies (e.g., Harel et al., 2016) suggest $n$ isn't necessarily positive. Although this is described later in the manuscript, the use of 'positive' here is misleading.

Line 88: What is 'steady-state' here? With no uplift rate imposed, it would imply $E = 0$; how does this fit into $k_s$ and $\theta$? $U$ also implies base-level lowering, which can justify the use of Eqs. (4) – (6), but I think some elaboration is needed.

Line 97 – 101: Although Supplementary Information describes the extraction of eroded volumes and incision, there's a fair bit of methodology that is lacking and should be explained (see comments below). Also, I have looked through the data repository, although much of the data is present, the 'ad hoc' Matlab scripts for analysis appears missing.

Lines 109 – 111: I'd recommend different abbreviations for maximum surface slope and surface slope mode. MSS and SSM are difficult to keep track of, maybe use $SS_{Max}$ and $SS_{Mode}$ to help keep these terms clear. Furthermore, these are two metrics that need clarified – How are these metrics calculated? Are there regions of the DEM that were ignored for MSS & SSM?

Fig. 1a-b: Going along with my comment above, I don't understand how box plots are being created for MSS and SSM. Max and mode would imply a single value for each model time step, not a distribution. Are these taken from the final time step through a moving window (similar to MLR/LRM)? Are these the values of the 10 representative time steps (and if so, does it make sense to have these since these values are evolving towards one value that may not overlap with the median implied by the plots)?

Furthermore, how much of the trend in Fig. 1a is simply a product of the initial condition, as opposed to evolution of the model? If these values were normalized by the original slope, would these trends still exist?

Line 117: Is there a reason that Figs. S2/S3 are kept in the supplement? Fig. S2 is the main observational item for how the models evolve, and provides the foundational data that Figs 1-4 are built, shouldn't it be in the main text?

Line 120: Again, some clarity is needed in the methodology (either in main text or supplement). How are channels being determined? Is there an imposed drainage area threshold? Many of these models appear to have channel widths that are multiple pixel sizes, how is that being accounted for with TopoToolbox? Are $k_s$ and $\theta$ determined by power-law regression between local slope and area?

Line 124: Similar to before, I recommend using different abbreviations for maximum local relief and local relief mode.

Line 186: What is $V_n$ normalized by? What is its normalized range and units? Normalization implies that it should be unitless, which would make $Ae$ dimensional. Or is $V_n$ the value used to normalize $V$? How is incision rate calculated here – is it the mean or max rate over the entire domain?

Line 194: This statement (and Fig. 2.c-d) suggest these trends may be better represented with a log-scale on the y-axis?

Lines 195 – 197 and Fig. 2d: I question the use of plotting $S_e/R$ against $Ae$. Since $Ae$ contains $S_e/R$, it would be expected that these values relate. Would it be more beneficial to plot $S_e/R$ against $IV_n/V$ to more clearly show the relationship between forcing and response?

Lines 201 – 203 and labels of Fig. 2d: This description and figure labels seem at odds with the data and earlier discussion. Topography of mod2070 shows incised channels (Fig. S2), yet Fig. 2d's characterization is of a no-channel landscape. The authors qualify this as incision being broadly distributed; however, earlier they highlighted mod2070 as an example of high regional slopes being able to overcome high precipitation rates and have enough potential energy for channelization (lines 160 – 162). I don't see how mod2070 can be used as evidence for both the trade-off between slope and precipitation to form channels, and having erosion too-broadly distributed for channels to be relevant?

Lines 211 – 213: Given the concerns above, I don't see how $Ae$ in its current form provides enough information to be used a threshold parameter or some predicting factor. In its current formulation, evaluating $Ae$ requires an analog model to already be ran, so it would be apparent whether channels develop. $S_e/R$ alone seems a better predictor than $Ae$.

Line 242 and Fig. 4: I assume these $k_{sn}$ values were all normalized by the same concavity index? This should be stated somewhere (and that value that was used).

**Technical Corrections:**

Line 119: I think a small typo – 'the final stage'.

Line 134: I think magnitude is missing from the stated value – should it be 1.4 x10$^6$ mm$^3$?

Lines 195, 198, 204, 207: Fig. 1d or Fig. 2d?

---

## Author Comment (AC1)

Riccardo Reitano Univ. Roma Tre

To: Dr. Prof. Greg Hancock Editor, *Earth Surface Dynamics*

Reference: Response to referees' comments and marked-up manuscript version showing the changes made regarding the submission *"Stream laws in tectonic landscape analogue models"* [EGUSPHERE-2022-911] by R. Reitano, R. Clementucci, E. M. Conrad, F. Corbi, R. Lanari, C. Faccenna, C. Bazzucchi.

Dear Dr. Prof. Greg Hancock,

first, we would like to thank the reviewer and yourself for the corrections and the useful comments. We include hereafter, point by point, the reply (in *dark red italic* text) to all the reviewers' comments. We will also upload the revised version of our manuscript with changes in a blue font. We hope the manuscript is now ready for publication in *Earth Surface Dynamics*.

With my kindest regards,

Riccardo Reitano, on behalf of all authors.

**RESPONSE TO REFEREE COMMENTS**

**Reviewer #1**

Line 15 - 16: Authors argue for the need to adequately model tectonic settings, but their setup simulates low uplift rates / passive margins. This motivation seems slightly skewed.

We agree that the motivation may be not clear in the first lines of the abstract. We modified the abstract (lines 3-6).

Line 48: Here and elsewhere in the paper, 'boundary condition' seems to be used interchangeably with model forcing; these are not the same. Rainfall rate and imposed slope should be described only as model forcings (and even then, the imposed slope is more of an initial condition).

In the jargon of analogue modelling, boundary conditions commonly represent all the a priori imposed conditions: geometries, velocities, mechanical properties etc. constraining the evolution of a model (e.g., Schellart et al., 2016, Corbi et al., 2016; Corbi et al., 2017a,b; Funiciello et al., 2004; Graveleau et al., 2011, 2012; Rosenau et al., 2017). Usually only materials and model set-up (e.g., extensional, or compressional apparatus) may fall outside the definition of boundary conditions (Schreurs et al., 2006, 2016). That's the reason why we will maintain this phrasing/concept over the whole revised version of the paper.

Line 63: Authors don't change nozzle flow rate to test precipitation rates, but instead change the number of nozzles. This should be elaborated on – what configurations were used and what tests conducted to ensure uniform rainfall?

We modified the text accordingly (lines 43-44), citing a previous work where the procedure is explained in detail.

Line 84: Some studies (e.g., Harel et al., 2016) suggest n isn't necessarily positive. Although this is described later in the manuscript, the use of 'positive' here is misleading.

We agree with the reviewer and changed the text accordingly (lines 64).

Line 88: What is 'steady-state' here? With no uplift rate imposed, it would imply E = 0; how does this fit into  $k_s$  and  $\theta$ ? *U* also implies base-level lowering, which can justify the use of Eqs. (4) – (6), but I think some elaboration is needed.

The sentence was wrongly phrased. We agree with the reviewer that, in the absence of U, is wrong to speak about "steady-state". Nevertheless, we still can describe the slope-area relationship through the Flint's Law (lines 68-69).

Line 97 - 101: Although Supplementary Information describes the extraction of eroded volumes and incision, there's a fair bit of methodology that is lacking and should be explained (see comments below). Also, I have looked through the data repository, although much of the data is present, the 'ad hoc' Matlab scripts for analysis appears missing.

We modified the main text and Supplementary material, "Eroded volumes" section. We decided not to upload the Matlab scripts since we only used functions from TopoToolbox, without modifying them and already open-access and downloadable from the website. Still, we forgot to change it in the main text. We modified the main text accordingly.

Lines 109 - 111: I'd recommend different abbreviations for maximum surface slope and surface slope mode. MSS and SSM are difficult to keep track of, maybe use  $SS_{Max}$  and  $SS_{Mode}$  to help keep these terms clear. Furthermore, these are two metrics that need clarified – How are these metrics calculated? Are there regions of the DEM that were ignored for MSS & SSM?

We agree with the reviewer's suggestion to rename the metrics. We also added some lines on how the metrics are calculated (lines 86-93). It should now be clear how we generated the boxplots (comment below).

Fig. 1a-b: Going along with my comment above, I don't understand how box plots are being created for MSS and SSM. Max and mode would imply a single value for each model time step, not a distribution. Are these taken from the final time step through a moving window (similar to MLR/LRM)? Are these the values of the 10 representative time steps (and if so, does it make sense to have these since these values are evolving towards one value that may not overlap with the median implied by the plots)?

Part of this comment was answered in the previous one. What it is correctly pointed out by the reviewer is

"does it make sense to have these since these values are evolving towards one value that may not overlap with the median implied by the plots)?"

The values are correctly changing following the model evolution. Nevertheless, as highlighted in the manuscript, the overall behavior of the model is function of the applied imposed slope and rainfall rate. This suggests (and is quantified and showed in the manuscript) that even if the model is evolving through time, the values for SSmax, SSmode, LRmax and LRmode are grouped into a varibility that depends mainly on the applied boundary conditions. It is very clear in models with very high rainfall rate, where the distribution of data strongly differs from models with lower rainfall rate (higher distributions but over lower values). The idea to use boxplots to display the data was that the variability of every basin at every time step still reflects the applied boundary conditions.

Furthermore, how much of the trend in Fig. 1a is simply a product of the initial condition, as opposed to evolution of the model? If these values were normalized by the original slope, would these trends still exist?

We agree with the reviewer. As presented in the manuscript, the effect of the imposed slope was not explained. We add a description (lines 91-93) about how imposed slope controls the  $SS_{max}$ . Normalizing the values would probably result in no trends for the  $SS_{max}$  between models 1009-1509-2009, 1022-1522-2022, and 1070-1570-2070. Nevertheless, we decided to avoid normalization of the dataset because the regional slopes reflect the different topographic surfaces and hillslope that characterize the large variety of slow tectonic settings. Moreover, with this figure we also aim at showing how increasing the rainfall rate increases the  $SS_{max}$  and how this behavior is opposite with respect to  $SS_{mode}$  within the same imposed regional slope. This is the reason why we arranged the figure this way. We hope that the new lines will clarify this point.

Line 117: Is there a reason that Figs. S2/S3 are kept in the supplement? Fig. S2 is the main observational item for how the models evolve, and provides the foundational data that Figs 1-4 are built, shouldn't it be in the main text?

The manuscript was originally submitted as a "short communication", later changed by the editor. This is the reason why Figs. S2/S3 were in the supplement. Since the manuscript is no more a "short communication", we agree with the Reviewer on moving these figures in the main text (Fig. 2).

Line 120: Again, some clarity is needed in the methodology (either in main text or supplement). How are channels being determined? Is there an imposed drainage area threshold? Many of these models appear to have channel widths that are multiple pixel sizes, how is that being accounted for with TopoToolbox? Are  $k_s$  and  $\theta$  determined by power-law regression between local slope and area?

We agree with the reviewer. We describe how channels are calculated and how ks and  $\theta$  are computed in the main text and in the supplementary (lines 77-78). For the channel extraction we used, channels widths are not considered, as we now explain in the supplement.

Line 124: Similar to before, I recommend using different abbreviations for maximum local relief and local relief mode.

Modified accordingly with the Reviewer.

Line 186: What is  $V_n$  normalized by? What is its normalized range and units? Normalization implies that it should be unitless, which would make Ae dimensional. Or is  $V_n$  the value used to normalize V? How is incision rate calculated here – is it the mean or max rate over the entire domain?

Lines 195 – 197 and Fig. 2d: I question the use of plotting  $S_e/R$  against *Ae*. Since *Ae* contains  $S_e/R$ , it would be expected that these values relate. Would it be more beneficial to plot  $S_e/R$  against  $IV_n/V$  to more clearly show the relationship between forcing and response?

Lines 211 - 213: Given the concerns above, I don't see how Ae in its current form provides enough information to be used a threshold parameter or some predicting factor. In its current formulation, evaluating Ae requires an analog model to already be ran, so it would be apparent whether channels develop. Se/R alone seems a better predictor than Ae.

We did not modify the text following the two first comments since we modify the text and the figure accordingly with the last comment ("Given the concerns above [...]"). We do not use Ae anymore, using only Se/R as predictor (lines 156-177).

Line 194: This statement (and Fig. 2.c-d) suggest these trends may be better represented with a log-scale on the y-axis?

We modified the new Fig. 5 accordingly.

Lines 201 - 203 and labels of Fig. 2d: This description and figure labels seem at odds with the data and earlier discussion. Topography of mod2070 shows incised channels (Fig. S2), yet Fig. 2d's characterization is of a no-channel landscape. The authors qualify this as incision being broadly distributed; however, earlier they highlighted mod2070 as an example of high regional slopes being able to overcome high precipitation rates and have enough potential energy for channelization (lines 160 – 162). I don't see how mod2070 can be used as evidence for both the trade-off between slope and precipitation to form channels, and having erosion too-broadly distributed for channels to be relevant?

We agree with the reviewer. The previous motivations were unclear. We rephrase the sentence (lines 174-177) to highlight the behavior of model 2070. We also modified the figure (new Fig. 5).

Line 242 and Fig. 4: I assume these  $k_{sn}$  values were all normalized by the same concavity index? This should be stated somewhere (and that value that was used).

We agree with the Reviewer. We forgot to add this information (lines 208).

We implemented all the technical corrections.

**Reviewer #2 – Greg Hancock**

An issue that needs to be better addressed is that of the relevance of scale of the experiments and the relevance to the real world scale. This needs to be better discussed both in the discussion as well as the Conclusion.

We agree with the Reviewer and we added lines both in Discussions and Conclusion (lines 182-184, 230-231).

It is also not clear what is meant by the suggestion of diffusion - 'Higher rainfall rates (70 mm h-1 in this work) tend to inhibit the development of a channelized and branching channel network in favor of more diffusive and mass wasting processes. This trend can be deduced by analyzing the DEMs of mod1070 and mod1570 (Fig. S2), or simply by noting the diffuse nature of erosion under high rainfall conditions (Fig. S3)'. It is not clear what diffusion occurs as the raindrops are small

We added few lines that should clarify more what we mean by "diffusion" in the manuscript. Here diffusion is not function of the raindrops, but is a function of the capacity of the system of channelizing water. Geometrically speaking, water does not collect into straight lines, but flow as a water-sheet allowing water runoff (Viaplana-Muzas et al., 2015) eroding greater areas (line 140-141), but with lower incision. We added this information to the manuscript (lines 128-131).

Please explain why such a large rainfall range was used and how this scales to real world application.

*Please see the first comment of Reviewer #2.*

**References**

- Graveleau, F., & Dominguez S. (2008). Analogue modelling of the interaction between tectonics, erosion, and sedimentation in foreland thrust belts. *Comptes Rendus Geoscience*, *340*, 324–333, https://dx.doi.org/10.1016/j.crte.2008.01.005
- Graveleau, F., Hurtrez, J.-E., Dominguez, S., & Malavieille, J. (2011). A new experimental material for modeling relief dynamics and interactions between tectonics and surface processes. *Tectonophysics*, *513*(1–4), 68–87. https://doi.org/10.1016/j.tecto.2011.09.029
- Graveleau, F., Strak, V., Dominguez, S., Malavieille, J., Chatton, M., Manighetti, I., & Petit, C. (2015). Experimental modelling of tectonics-erosion-sedimentation interactions in compressional, extensional, and strike-slip settings. *Geomorphology*, 244, 146–168. https://doi.org/10.1016/j.geomorph.2015.02.011
- Viaplana-muzas, M., Babault, J., Dominguez, S., Driessche, J. Van Den, & Legrand, X. (2015). Tectonophysics Drainage network evolution and patterns of sedimentation in an experimental wedge. *Tectonophysics*, 664, 109–124. https://doi.org/10.1016/j.tecto.2015.09.007

---

## Author Response (AR2)

**Riccardo Reitano**
**Univ. Roma Tre**

To: Dr. Prof. Greg Hancock
Editor, *Earth Surface Dynamics*

Reference: Response to minor revisions made by the editor

Dear Dr. Prof. Greg Hancock,

We provide a new version of the manuscript, where we checked for any spelling and grammar issues. Since we performed minor corrections, the manuscript and its marked-up version correspond in terms of colors, highlights etc.
Thank you for your effort and the useful suggestions.

With my kindest regards,

Riccardo Reitano, on behalf of all authors.